# Addressing Linguistic Bias through a Contrastive Analysis of Academic Writing in the NLP Domain

**Robert Ridley**    **Zhen Wu**[*]    **Jianbing Zhang**    **Shujian Huang**    **Xinyu Dai**

National Key Laboratory for Novel Software Technology, Nanjing University, Nanjing, China
Collaborative Innovation Center of Novel Software Technology and Industrialization, China
robertr@smail.nju.edu.cn
{wuz,zjb,huangsj,daixinyu}@nju.edu.cn

## Abstract

It has been well documented that a reviewer's opinion of the nativeness of expression in an academic paper affects the likelihood of it being accepted for publication. Previous works have also shone a light on the stress and anxiety authors who are non-native English speakers experience when attempting to publish in international venues. We explore how this might be a concern in the field of Natural Language Processing (NLP) through conducting a comprehensive statistical analysis of NLP paper abstracts, identifying how authors of different linguistic backgrounds differ in the lexical, morphological, syntactic and cohesive aspects of their writing. Through our analysis, we identify that there are a number of characteristics that are highly variable across the different corpora examined in this paper. This indicates potential for the presence of linguistic bias. Therefore, we outline a set of recommendations to publishers of academic journals and conferences regarding their guidelines and resources for prospective authors in order to help enhance inclusivity and fairness.

## 1 Introduction

Artificial Intelligence (AI) research has experienced a boom in recent years, seeing a significant increase in published works from across the globe. With the international language of academia being English, a significant number of publications are written by non-native speakers of English. This phenomenon brings about the issue of linguistic bias (Politzer-Ahles et al., 2016; Hanauer et al., 2019; Flowerdew, 2019; Politzer-Ahles et al., 2020; Soler, 2021).

Described by Politzer-Ahles et al. (2020), linguistic bias, also known as linguistic injustice, is the case of academic writing being judged more harshly by reviewers and editors if it doesn't meet the standards of international academic English,

even if the quality of the content is sufficient and communicability isn't impacted. This is supported by Strauss (2019); Yen and Hung (2019), who conclude that reviewers are more likely to accept papers whose language is more nativelike.

To make steps towards addressing the issue of linguistic bias, we conduct a comprehensive contrastive statistical analysis of paper abstracts from the NLP domain, where we investigate how writing differs between authors from different linguistic backgrounds. We choose this domain due to its value in the NLP community, where computational methods are used to study language and linguistic expression is seen as important.

In our analyses, we explore aspects of *Organizational Competence*, as outlined by Bachman et al. (1990). More specifically, we focus on the sub-components of *Grammatical Competence* and *Textual Competence*, within which we analyse how **lexical**, **morphological**, **syntactic** and **cohesion** usage varies for authors from different geographical locations. Through our analyses, we identify a number of characteristics that vary widely between writers from different linguistic backgrounds. For instance, we identify that in the *China, Japan and India corpora from our dataset, there are varying preferences for specific lexical bundles*, and *all corpora in our dataset have vast differences in the discourse connectors that are preferred*.

Therefore, to support writers whose usage patterns may impact the expression of the ideas presented in their work and enhance their chances of being evaluated fairly in international academic publications, we also outline a set of recommendations for conferences and journals regarding the author guidelines, resources and tools they make available.

In summary, the contributions of our work are as follows:

- To explore how linguistic bias might be present in NLP, we are the first to perform

---

*Corresponding author

a comprehensive contrastive analysis of academic NLP writing from authors of different native-language backgrounds.

- Through our analysis, we identify how writers of different nationalities differ in their expression across the dimensions of lexis, morphology, syntax and cohesion, and conclude that there are many aspects of writing that may give rise to linguistic bias.

- We determine that action is required to address the issue of linguistic bias, and detail a set of recommendations to help alleviate the issue.

## 2 Related Work

Some of the earliest work related to differences in language output based on one's native language comes from Lado (1957), who posited that features of the target language that differed greatly from a learner's native language would be more difficult for the learner to learn. Conversely, features of the target language that are similar to a learner's native language would be easier for the learner to learn.

More recently, with the support of learner corpora, such as Granger et al. (2020), researchers have sought to identify how writers of different native languages differ in their writing when compared with native English speakers. We summarize findings of works focusing on *lexis*, *morphology*, *syntax* and *cohesion* below.

**Lexis**    Existing works on lexis have identified that measures of lexical diversity, sophistication and density are higher for non-native English-speaking students with higher proficiency (Memari, 2021; Prados, 2010).

Non-native English speakers also tend to use more lexical bundles and use them more frequently than native English speakers (Hyland, 2008a; Wei and Lei, 2011; Pan et al., 2016; Bychkovska and Lee, 2017) in order to produce text that appears more native-like.

**Morphology**    It has been found that it is possible for morphological complexity to reach native-like levels for second-language speakers (Brezina and Pallotti, 2019). However, it is common for complexity to be lower for those whose native language is less morphologically complex than the target language, as they may have yet to acquire the target structure.

**Syntax**    Comparisons of syntax are typically made through the lens of complexity at different levels, such as phrasal, clausal and sentence. Previous works have identified developmental stages for syntactic complexity in second-language learners of English. For example, Biber et al. (2011) proposed that complement clauses are developed first and then phrasal modifiers are acquired later. Other research, however, has found that these developmental stages are not always observed. For instance, Lu and Ai (2015) recognized that Chinese, Japanese and Russian writers use far less subordination than both native English speakers and other non-native English speakers. They also recognized that Chinese writers use fewer coordinating clauses than writers from other backgrounds. The authors suggested that these observations are a result of native-language influence.

**Cohesion**    It is common for researchers to compare the usage of discourse connectors of different groups of writers. For instance, Milton and Tsang (1993) compared native English speakers and Cantonese speakers in their use of 25 connectors in the categories of *additive*, *adversative*, *causal* and *sequential*. They found that the Cantonese speakers had a considerably higher usage for a number of connectors such as *firstly*, *secondly*, *lastly*, *besides*, *moreover* and *therefore*; and lower usage for connectors such as *likewise* and *previously*.

Existing works on contrastive analysis in a second-language setting generally focus on analysing writing by language learners or undergraduate students. Moreover, they are limited in their scope, typically focusing on one area of language — e.g., syntax (Wong and Dras, 2009) or cohesion (Milton and Tsang, 1993). This study differs from previous research in that it is the first to perform a contrastive analysis of professional academic writing in the NLP domain. Thus, we are able to analyze characteristics specific to academic writing in our target domain and can give more relevant recommendations. A further point of difference is that we carry out a more comprehensive analysis than previous research, focusing on a wider range of language features.

## 3 Data Collection

Our data[1] are acquired from the following academic NLP venues: ACL 2018–2021, EMNLP

---

[1] Our data and analysis code are available at `https://github.com/robert1ridley/linguisticBias`

| Location | Abstracts | | | | Word Count | | | |
|---|---|---|---|---|---|---|---|---|
| | ACL | EMNLP | arXiv | All Data | ACL | EMNLP | arXiv | All Data |
| China | 487 | 646 | 2,442 | **3,575** | 69,286 | 92,755 | 389,081 | **551,122** |
| United States | 257 | 356 | 1,492 | **2,105** | 34,269 | 49,303 | 226,036 | **309,608** |
| Germany | 106 | 124 | 603 | **833** | 13,877 | 16,692 | 89,555 | **120,124** |
| Japan | 90 | 108 | 305 | **503** | 10,857 | 13,853 | 44,957 | **69,667** |
| India | 59 | 55 | 338 | **452** | 8,753 | 7,193 | 53,839 | **69,785** |
| **Total** | 999 | 1,289 | 5,180 | **7,468** | 137,042 | 179,796 | 803,468 | **1,120,306** |

Table 1: Dataset Statistics (punctuation removed)

2017–2021 and 'Computation and Language' tagged papers from arXiv.org published between January 1 2015 and April 24 2022. After collecting abstracts from these venues, we apply a set of heuristics (detailed in Appendix A) to determine the nationality of the authors.

In order to ensure we have a sufficient quantity of text on which to perform our analyses, we limit our analyses to the five most common geographic locations in the dataset. An overview of our data is shown in Table 1.

## 4 Analyses

As outlined in Section 1, we perform contrastive analyses across the lexical, morphological, syntactic, and cohesion dimensions of the different geographic locations in our dataset.

In summary, we find the following:

**Lexis** In a number of corpora in our dataset, synonyms, hypernyms and hyponyms are used at a lower rate, resulting in lower lexical complexity. In addition, in these corpora there is an increase in usage of fixed lexical expressions. This is evident in the China, India and Japan corpora, for example, where the usage of fixed lexical expressions is roughly double that of the United States corpus.

**Morphology** The distributions of verb tense usage are generally similar across the corpora. However, we notice that in the Japan corpus past tense verbs are used far more frequently and non-third person singular present verbs far less frequently than in all other corpora.

**Syntax** The United States corpus typically exhibits less syntactic complexity than all other corpora at phrase level. However, it is found that there is more complexity at clause level and sentence level.

**Cohesion** In the United States corpus, there is a greater usage of discourse connectors. However, each of the other corpora has its own specific set

| Location | Token-Type Ratio | Chain Length |
|---|---|---|
| China | 0.61* | 1.36* |
| United States | 0.63 | 1.39 |
| Germany | 0.63 | 1.36* |
| Japan | 0.61* | 1.33* |
| India | 0.61* | 1.34* |

Table 2: Token-Type Ratio (calculated as unique tokens divided by total tokens) and Average Lexical Chain Length. A pairwise t-test is carried out for the United States corpus against each of the other corpora. With the p-value set to 0.05, statistically significant results are marked with *.

of connectors that are used to a much higher extent than in any other corpus.

The details of our analyses and findings are outlined in the following subsections.

### 4.1 Lexical

We analyze the lexical characteristics of each location at two different levels. First, we look at lexical diversity at token level through calculating the token-type ratio. Then, we explore lexis at a multi-word level through a lexical-bundle analysis, where we identify commonly used lexical chunks.

#### 4.1.1 Lexical Diversity

**Analysis Methodology** To inspect lexical diversity, we aim to capture the proportion of unique words in each abstract. If there is a higher proportion of unique words, this will indicate that a larger range of vocabulary is used and thus a higher degree of lexical diversity. We measure this through calculating each abstract's token-type ratio, which is carried out by dividing the number of unique words by the number of total words.

In addition, we also aim to quantify the usage of synonyms, hypernyms and hyponyms. To do this we perform a lexical chain analysis. First, we extract all nouns from each abstract. Then, utilizing the WordNet interface provided by the python NLTK package[2], words that are considered

---
[2]https://www.nltk.org/

| Location | Bundles per MW | Unique Bundles | NP | PP | VP | Clause | Conj | Other |
|---|---|---|---|---|---|---|---|---|
| China | 10,422.37 | 50 | 9 | 12 | 8 | 19 | 1 | 1 |
| United States | 4,521.85 | 23 | 3 | 5 | 2 | 10 | 2 | 1 |
| Germany | 5,527.62 | 32 | 4 | 10 | 5 | 12 | 1 | 0 |
| Japan | 9,344.45 | 47 | 5 | 10 | 6 | 21 | 1 | 1 |
| India | 8,999.07 | 54 | 13 | 19 | 4 | 16 | 1 | 1 |

Table 3: Four-word Bundle Statistics: *Bundles per MW* refers to the number of four-word lexical bundles per million words, *Unique Bundles* to the number of unique bundles, *NP* noun phrase-based bundles, *PP* preposition phrase-based bundles, *VP* verb phrase-based bundles, *Clause* clause-based bundles, and *Conj* conjunctions.

| Bundle | China | United States | Germany | India | Japan |
|---|---|---|---|---|---|
| in this paper we | 2866.88 | 1343.63 | 1215.41 | 2221.11 | 1794.25 |
| in this work we | 593.34 | 807.47 | 907.4 | 673.5 | 315.79 |
| this paper proposes a | 143.34 | 83.98 | - | - | 272.73 |
| this paper presents a | 79.84 | - | - | 71.65 | 86.12 |
| in this study we | - | 83.98 | 83.25 | - | 602.87 |
| this study focuses on | - | - | - | - | 114.83 |
| this paper describes our | - | - | - | 186.29 | 86.12 |
| this paper describes the | - | - | - | 143.3 | - |
| **Total** | 3683.4 | 2319.06 | 2206.6 | 3295.85 | 3272.71 |

Table 4: Bundle counts (per million words) for function of introducing main idea of paper.

synonyms, hypernyms or hyponyms are combined into the same lexical chain. Once these lexical chains have been generated, we calculate the average chain length. Longer chains indicate more usage of synonyms, hypernyms and hyponyms.

**Findings** The results of our analysis are displayed in Table 2. For results from each of the publication venues in the dataset, we refer the reader to Table 9 in Appendix C.1.1. From looking at the *Token-Type Ratio* column, we can see that the United States and Germany corpora have the highest ratios, which indicates a higher degree of lexical diversity. We suggest that this is a result of increased usage of synonyms, hypernyms and hyponyms. This is supported by our lexical chain analysis, which is displayed in the *Chain Length* column of Table 2. The average lexical chain length in the United States corpus is the longest of all corpora, meaning that on average there is the use of a wider range of vocabulary to describe similar concepts. In contrast, the Japan and India corpora have the shortest average chain lengths, meaning that a decreased use of semantically-related terms is a likely factor in their lower token-type ratio scores.

### 4.1.2 Lexical Bundles

**Analysis Methodology** We perform a lexical-bundle analysis with the goal of capturing commonly used lexical sequences for different groups of writers. In our analysis, we collect all instances of four-word bundles that meet the criteria of occurring 20 times per million words in the corpus and appearing in at least one percent of abstracts for the specific corpus. A description regarding our selection of these criteria and additional processing is covered in Appendix B.

Table 3 shows the normalized bundle counts (number of bundles per million words) for the different geographic locations in our dataset. It also displays the total number of unique bundles, with each unique bundle being classified according to whether it is noun phrase-based, preposition phrase-based, verb phrase-based, clause-based, a conjunction or other. In classifying each bundle, we use the categories provided by Lu and Deng (2019); we additionally add a category *Other* for bundles that do not fit the predefined categories. For explanations of each category and corresponding examples, please refer to Table 8 in Appendix B.

**Findings** From looking at the number of bundles per million words in Table 3, we can see that in three corpora —China, Japan and India —lexical bundle usage is more than double that of the United States corpus. Additionally, while not to the same degree, the Germany corpus also exhibits higher lexical bundle usage than the United States corpus, with an increase of over 1,000 more bundles per million words. As can also be observed, not only are bundles used more frequently in the corpora from non-native English speaking locations, but the range of bundles used is also higher. We can see that the number of unique bundles for the China, Japan and India corpora (*Unique Bundles* column in Table 3) are all at least twice that of the United

States corpus.

Our findings are consistent with a number of previous lexical-bundle analysis studies, which observe that writers using their second language use lexical bundles more frequently than writers who are writing in their native language (Hyland, 2008a; Wei and Lei, 2011; Pan et al., 2016; Bychkovska and Lee, 2017). As described in Bychkovska and Lee (2017), writers using their second language rely on using fixed expressions in order to help them produce more academic-like texts and to avoid producing expressions that could be perceived as unconventional. We examine this phenomenon more closely by investigating bundle usage for a specific function within the abstract, which is that of introducing the main idea of the paper. The normalized bundle counts for this function are displayed in Table 4.

Here, it is apparent that in the China, Japan and India corpora, there is a higher reliance on using fixed expressions to carry out the function of introducing the main idea of the paper, with the total number of bundles used by these nationalities being between 41% (for the Japan corpus) and 69% (for the China corpus) greater than in the United States corpus. Interestingly, however, these three corpora differ vastly in the bundles they use. For instance, the China corpus has the highest proportion of bundle counts for this function, but these are distributed across only four different bundle types (*in this paper we*, *in this work we*, *this paper proposes a* and *this paper presents a*). From looking at the results, the high use of the single bundle *in this paper we* seems to be the main reason why bundle usage is high for this particular function, with usage being more than double that of the United States corpus.

While the Japan and India corpora both have a preference for using the bundle *in this paper we*, it is to a lesser extent than in the China corpus. In contrast to the China corpus, the high bundle use in these two corpora appears to be as a result of using a wider range of bundles, with both corpora recording usages of five and seven bundles respectively. Moreover, through observing the types of bundles used, we can also identify differences in terms of lexical preferences. For example, in the Japan corpus, there is a preference for the use of *study* when compared to the other corpora. The bundle *in this study we* is used over 600 times per million words in the Japan corpus, more than 6

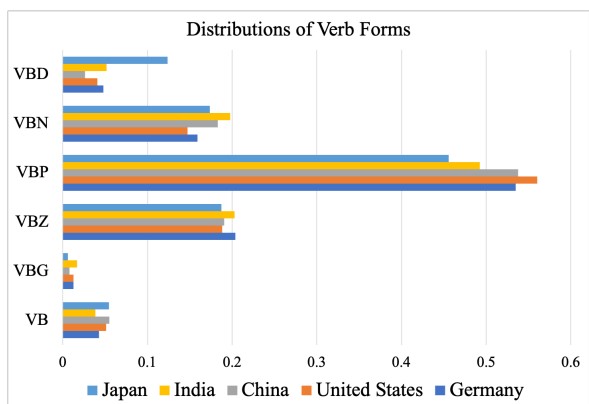

Figure 1: Distribution of Verb Forms for each corpus: *VBD* denotes past tense form (e.g., *they thought*), *VBN* past participle (e.g., *a sunken ship*), *VB* a verb's base form (e.g., *think*), *VBZ* third-person singular present (e.g., *she thinks*), *VBP* non-third person singular present (e.g., *I think*) and *VBG* gerund (e.g., *thinking is fun*).

times more frequently than any other corpus. Likewise, it is the only corpus that registers usages for *this study focuses on*.

In contrast, the Germany corpus features less bundle usage for this particular function, with total usage slightly below that of the United States corpus. This likely means that on average there is less usage of fixed expressions in order to carry out the function of introducing the main idea of a paper.

## 4.2 Morphological

**Analysis Methodology** To analyze the morphological dimension, we investigate the distributions of different verb forms by writers from each geographic location. Specifically, for each sentence in the respective corpus, we identify the main verb and classify[3] the verb form according to whether it is past tense, past participle, base form, third-person present, non-third person present or a gerund. The distribution results are displayed in Figure 1. For results of each publication venue in the dataset, we refer the reader to Figure 2 in Appendix C.1.2

**Findings** One key point of salience that can be observed is that in the Japan corpus the past tense is used significantly more frequently than in each of the other corpora in our dataset; 12.4% of all main verbs in the Japan corpus are in the past tense form. This is more than double that of the India corpus (5.2%), which is the next highest. The increased use of the past tense in the Japan corpus appears

---

[3]We utilize the python package spaCy: https://spacy.io/

to be as a result of a decreased use of non-third person present tense (only 45.5% in comparison to 56% in the United States corpus).

### 4.3 Syntactic

**Analysis Methodology** In our analyses of syntax, we explore the complexity at phrase-level, clause-level and sentence-level. To do this, we employ a variety of measurements: the number of noun-phrase modifiers, the number of clauses per sentence, the average parse-tree depth, and the average sentence length.

| Location | NP Mods | Cl per Sent | PT Depth | Sent Length |
|---|---|---|---|---|
| China | 1.75* | 1.88* | 6.24* | 24.56* |
| United States | 1.62 | 2.04 | 6.82 | 26.33 |
| Germany | 1.63 | 1.88* | 6.64* | 25.06* |
| Japan | 1.67* | 1.92* | 6.51* | 24.64* |
| India | 1.70* | 1.69* | 6.62* | 24.03* |

Table 5: Syntactic Complexity Measures (*NP Mods* is the average number of noun-phrase modifiers per noun phrase, *Cl per Sent* is the average number of subordinating clauses per sentence, *PT Depth* is the average parse tree depth for a sentence, and *Sent Length* is the average number of tokens per sentence). Pairwise t-tests are used to compare the United States corpus with all other corpora. Statistically significant results with a p-value of 0.05 are notated with an asterisk (*).

**Findings** The results of our analyses are displayed in Table 5. Results across each publication venue in the dataset can be found in Table 10 from Appendix C.1.3. In Table 5, we can see that the United States corpus has the fewest number of noun-phrase modifiers on average. In contrast, the location with the highest phrasal complexity by this metric is China, which, along with India and Japan, possesses a significantly higher degree of phrasal complexity than the United States corpus.

Our findings have a lot in common with those of Lu and Ai (2015), who analyse papers written by English language learners of a variety of nationalities and find that Chinese writers exhibit higher degrees of phrasal complexity than they do clausal complexity. Likewise, they observe that Japanese writers also possess a lower degree of clausal complexity. However, in contrast to our findings, they find that Japanese writers express a lower level of phrasal complexity than native English speakers.

We explore the preference for phrasal complexity as opposed to clausal complexity in the Japan corpus in our dataset. An example of a sentence exhibiting high phrasal complexity is shown here:

> *This article describes an efficient training method for online streaming attention-based encoder-decoder (AED) automatic speech recognition (ASR) systems.*

The sentence contains a long noun phrase (highlighted in red). The head noun of this phrase is *systems*, which is modified by three phrases preceding it (*online streaming*, *attention-based encoder-decoder* and *automatic speech recognition*). This sentence could also be expressed through the use of subordinate clauses. For example, as follows:

> *This article describes an efficient training method for online streaming automatic speech recognition (ASR) systems based on attention-based encoder-decoder (AED) architectures.*

As we can see, re-writing the sentence in this way would decrease the phrasal complexity, as now the constituents are broken up into smaller, less complex phrases (highlighted in red and blue). The sentence length and clause counts have also been increased as a result. This appears to go some way to explaining why this corpus has slightly less complexity at sentence level and clausal level but more complexity at a phrasal level when compared with the United States corpus. When we look at the average parse-tree depth and sentence length measurements, we also observe overall sentence complexity to be higher in the United States corpus and lower for the three corpora with the highest measures of phrasal complexity. This indicates that the complexity at sentence level in the United States corpus is likely as a result of increased clausal complexity.

### 4.4 Cohesion

**Analysis Methodology** As with previous works which have explored cohesion in the writing of second-language speakers (Milton and Tsang, 1993; Narita et al., 2004; Field and Oi, 1992; Goldman and Murray, 1992), we analyse cohesion through the use of cohesive devices and discourse connectors. We record all uses of the connectors provided by Kalajahi et al. (2017). In this work, the authors provide a list of 632 discourse connectors, which are divided into eight categories and a further 17 sub-categories. Summaries of these categories and sub-categories can be found in Table 12 in Appendix D.

| Location | Connectors per Sentence |
|---|---|
| China | 1.89* |
| United States | 1.98 |
| Germany | 1.98 |
| Japan | 1.82* |
| India | 1.81* |

Table 6: Average number of discourse connectors. A pairwise t-test is carried out for the United States corpus against all other corpora. Statistically significant results are denoted with * when the p-value is set to 0.05.

| Location | Connector | Usage Ratio |
|---|---|---|
| China | *firstly* | 11.49 |
| | *besides* | 10.60 |
| | *usually* | 5.53 |
| | *meanwhile* | 5.18 |
| | *matching* | 3.45 |
| Japan | *on the basis of* | 12.25 |
| | *accordingly* | 8.91 |
| | *considering* | 4.45 |
| | *simultaneously* | 4.28 |
| | *hence* | 3.77 |
| India | *hence* | 7.18 |
| | *in comparison* | 4.44 |
| | *namely* | 4.11 |
| | *in all* | 3.85 |
| | *right* | 3.33 |
| Germany | *at the same time* | 6.87 |
| | *to this end* | 3.25 |
| | *mostly* | 3.15 |
| | *thereby* | 3.09 |
| | *hence* | 2.97 |

Table 7: Five discourse connectors with highest usage ratio when compared with United States corpus

For each corpus, we calculate the average number of discourse connectors per sentence. These results are summarized in Table 6. Results for each publication venue can be found in Table 11 in Appendix C.1.4. Additionally, to gain a better understanding of connector usage, in Table 7 we display the five connectors in each corpus with the highest ratio of usage in comparison to the United States corpus.

**Findings** From Table 6, we can see that discourse connectors are used more frequently per sentence in the United States and Germany corpora, with the other three corpora showing less frequent usages. There are some interesting characteristics of other languages that might give indication as to why fewer connectors are used. For instance, a number of works (Wang, 2011) have identified that in Chinese, connectors are used less frequently than in English. This is a result of English expressing cohesion in a more explicit manner than Chinese, which relies more on context to imply cohesion

between sentences. For example, Zhou and Xue (2012) discover that 82% of tokens in their Chinese Treebank exhibit implicit relations, as opposed to around 54% in the English PDTB 2.0.

We also recognised that there are a number of cases where certain connectors have high usage rates across the different corpora, illustrated by Table 7. For instance, in the China corpus, the connector with the highest usage ratio (in comparison to the United States corpus) is *firstly*, whose usage is 11.49 times more frequent than in the United States corpus.

The connector *firstly* is often used either to give an argument or introduce a point for non-sequential ideas. However, it appears that in the China corpus there is a preference for using it as a sequential connector. To examine this, we calculated how frequently *then* is used in the sentence following the use of *firstly*, as these two terms are used together to describe event sequences. In the China corpus, 33% of the time *firstly* appears, the next sentence contains the connector *then*. Here is an example:

> *Firstly, we extract syntactic indicators under the guidance of syntactic knowledge. Then we construct a neural network to [. . . ]*

In this case, the two contributions of the work are introduced as two sequential items. While the order in which the contributions were completed may be sequential, presenting them sequentially is a style choice more common in the China corpus than in others; in all other corpora (except for one instance in the United States corpus) there are no cases of *firstly* being used with *then*. This pattern in the China corpus is in contrast with how these sequential adverbs are used in the other corpora. For example, in the United States corpus, *firstly* is often used with *secondly* to introduce linked ideas, rather than to introduce ideas or actions as a sequence:

> *The purpose of this paper is two-fold; firstly, we propose a novel attention model [. . . ] Secondly, we study the interaction between attention and syntactic structures [. . . ]*

Another connector that has a high usage ratio in the China corpus is *besides*. We discovered that a common usage pattern for this connector in our dataset is for linking ideas either in a descriptive or

analytical context. This leads to *besides* being preferred over *in addition* or *additionally* in the China corpus, whereas *besides* appears far less frequently in other corpora of the dataset. Here is an example from the China corpus:

> *Results show that our models outperforms existing methods on multi-domain dialogue, giving the state-of-the-art in the literature.* *Besides, with little training data, we show its transferability by outperforming prior best model by 13.9% on average.*

In this short extract, the first and second sentences are describing two separate contributions of the research paper, rather than the second sentence being an additional point in an argument.

A connector with high usage ratios in the Japan, India and Germany corpora is *hence*. This seems to be as a result of each of these three corpora having a preference for *consequential* type connectors. We observe that these three corpora contain the highest proportion of *consequential* connectors. Moreover, there are a number of other consequential connectors that are used significantly more frequently than in the United States corpus. For example, *thereby* is the connector in the Germany corpus with the fourth highest ratio and has the sixth highest ratio in the Indian corpus; and *therefore* has the seventh highest ratio in the Germany corpus and the sixth highest in the Japan corpus.

## 5 Recommendations

In order to make progress on the issue of linguistic bias and based on the findings of our analyses, we outline a set of recommendations to academic journals and conferences regarding how their author guidelines could support prospective authors from around the globe. Our recommendations are focused on the four areas of analysis performed in this study and are summarized as follows:

**Lexis** We found variation in lexical diversity across each of the investigated corpora due to differences in the use of synonyms, hypernyms and hyponyms. We recommend that author guidelines include lists of commonly used terms in NLP research and include examples of suitable synonyms.

We also found that there was a higher use of lexical bundles in a number of the corpora studied. To help avoid over-reliance on lexical bundles, we propose that author guidelines include suggestions for

how different pragmatic functions of the paper can be expressed. For instance, we found that different corpora in our dataset had different preferences for bundle usage as a means for introducing a paper's main idea. The author guidelines should provide multiple examples for how authors can introduce the main idea of their work, in addition to examples for other common pragmatic functions.

**Morphology** As we have identified, there are different preferences for grammatical tense across the different corpora in our dataset. In academic NLP papers, there are a number of common functions expressed, such as introducing one's contributions, citing related work, describing methods, introducing experiments, discussing results, etc. Each function carries its own set of expectations regarding the grammatical tense to be used (for example, which tense to use when citing literature in a *Related Work* section). To aid writers regarding usage of grammatical tense, we suggest that clear guidelines accompanied by sufficient examples be provided regarding tense usage for specific functions.

**Syntax** From our analysis, we can see that in some of the corpora investigated, there is a preference for expressing complex points through complex phrase structures. Increased complexity with regards to phrasing can impact understanding and communicability. We suggest that author guidelines include clear examples of how to increase readability through splitting phrases containing multiple modifiers. Journals and conferences could also provide free access to automated writing tools capable of paraphrasing, thus providing authors with ideas on how to rewrite complex content.

**Cohesion** As we have shown in this study, the preference for different discourse connectors across many of the corpora investigated are highly variable. We recommend providing a list of common connectors and examples of how they can be used. This can also be accompanied by a list of frequently overused or misused connectors along with alternative options for connectors.

## 6 Conclusions and Future Work

This paper seeks to address the issue of linguistic bias in academic publishing. Our comprehensive contrastive analysis of academic writing in the NLP domain identifies a number of characteristics that are highly variable across writers from different

nationalities. These findings highlight the potential risk of linguistic bias. To mitigate this risk, we outline a set of recommendations to academic publishers about how their resources could better support writers in presenting their ideas and contributions and thus ensuring that their work is fairly evaluated.

Regarding possible directions for future work, we propose that more work be done to analyze the writing of a wider range of author backgrounds. Moreover, our analyses could provide a basis for integration into writing correction and suggestion tools. With current solutions typically focusing on providing corrections for grammatical or convention-based errors, there is scope to extend this to suggestions based on writing characteristics where expression may impact an author's chance of being assessed fairly, but not necessarily be incorrect grammatically or according to conventions.

## Limitations

In order to ensure that we have sufficient data for each of the geographic locations in our analysis, we limit our focus to the five most common locations in our dataset. For future research, there is value in extending the analysis to include writers from a wider range of backgrounds.

Moreover, while we perform analyses on a range of language features, our focus is on the grammatical and textual components of writing. Exploring other areas of language, such as sociolinguistic aspects (e.g., use of register, figures of speech, and cultural references) could also be explored in future work.

Finally, we also acknowledge that, while we attempt to identify the location of where the writers originate, we cannot determine the native language or the variety of the language spoken by each writer. For instance, in India, there are many languages spoken, and in China, while the predominant language is Mandarin, there are many varying dialects spoken across the nation.

## Ethics Statement

English is considered a global language and it is natural that different preferences and patterns of usage exist for speakers from different geographical locations. In this work, we do not consider any variation to be correct or incorrect, but it is conceivable that differences in expression may impact how ideas are understood and perceived in academic writing. As such, our work seeks to identify some of these differences and to provide suitable recommendations regarding author guidelines to assist writers with the expression of their ideas and thus to better enable inclusivity and fairness.

The findings and recommendations outlined Section 5 may be sensitive to some. Both in this section and throughout the paper, we are careful to present our analyses across the different corpora objectively, terming differences as variations in production, or preferences for different production patterns, rather than to emphasize or evaluate differences of proficiency between native English speakers and non-native English speakers.

Variations in English production can result from a number of factors, such as linguistic backgrounds, educational systems, or language acquisition opportunities. It is our firm belief that such variations should not be the cause for discrimination and not be the basis for determining the value or quality of academic contributions.

The analyses in this paper are conducted on abstracts of academic papers published in conferences and on a pre-print server. We ensure that all data used in our analysis were obtained legally and ethically. Regarding licensing for our dataset, all data from the ACL and EMNLP conferences are covered by the Creative Commons Attribution 4.0 International License, and the arXiv data are covered by a Creative Commons CC0 1.0 Universal Public Domain Dedication. In order to respect the rights and privacy of the authors in the dataset, we are careful to remove any personally identifiable information.

We recognize the importance of responsible and ethical conduct in AI research and will continue to prioritize these values in our work.

## Acknowledgements

We thank all anonymous reviewers for their feedback and suggestions. This work is supported by the National Natural Science Foundation of China (No. 61976114, 62206126 and 61936012).

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

## A  Appendix

### A.1  Data Acquisition: Determining Author Nationality

First, we attempt to determine the nation of the institution from which the paper is published. To do this, we extract the listed email address of the first author. Then, we check whether the email matches a listed university domain[4] and thus we can extract the nation. For emails addresses which are not matched, we extract the top-level domain and match it with a nationality if possible. Papers without email addresses or in instances where the nationality of the institution cannot be determined are discarded.

Once we have collected a list of papers for which we know the nationality of the publishing institution, we determine whether the first author originates from the same nation as the publishing institution. To do this, we utilize a name origin database[5]. If the nation of origin of author's family name is listed as the same that of the publishing institution, we assume the author to originate from that nation. All papers for which this criterion is not met are discarded. As a further post-processing step, since many papers are published on preprint servers before being submitted to conferences, we are careful to de-duplicate papers in our dataset.

It is not impossible that in some cases, native English speakers could appear as non-first authors for papers in the non-native English speaker corpora. However, as determined by the heuristics outlined above, we do not observe a high instance count for this being the case. In addition, it is generally considered that first authors are the main contributors to scientific papers and we thus make the assumption that they are the writers in our corpora. We thus believe that the small quantity of native English speaker non-first authors in non-native English speaker corpora coupled with first authors generally making the largest contributions will mean that only a limited amount of noise could be added by native English speakers in these corpora.

## B  Appendix

### B.1  Lexical Bundles: Bundle Processing

To capture meaningful lexical sequences, we need to consider three criteria, lexical-bundle size, frequency and dispersion (as specified in Bychkovska and Lee, 2017).

#### B.1.1  Bundle Size

For bundle size, we follow existing works (Biber et al., 2004; Hyland, 2008b; Byrd and Coxhead, 2010; Ädel and Erman, 2012; Pan et al., 2016; Bychkovska and Lee, 2017; Lu and Deng, 2019) and extract four-word lexical bundles. This is because, as described by Cortes (2004), three-word structures are often contained in four-word structures (for example, Cortes, 2004 notes that *as a result* is contained in the structure *as a result of* ). In addition, four-word bundles are far more common than five-word bundles, thus providing a wider range of structures to analyse.

#### B.1.2  Bundle Frequency

The frequency criterion allows us to capture lexical chunks that occur frequently enough to be representative of the target register. Following previous works (Biber et al., 2004; Hyland, 2008b; Byrd and Coxhead, 2010; Ädel and Erman, 2012; Pan et al., 2016; Bychkovska and Lee, 2017; Lu and Deng, 2019), which set frequency thresholds at between 20 and 40 occurrences per million words, we choose a threshold of 20 occurrences per million words. Initially, we experimented with a few cut-off frequencies. Through manual inspection, we generally found that setting the frequency to 20 per 1 million words was sufficient for preventing the capture of rare low-frequency bundles. At higher frequencies, we found that there was a limited difference in the bundles that were captured, as bundles that met our dispersion threshold (discussed in Appendix Section B.1.3) typically occurred more frequently than 40 instances per million words.

#### B.1.3  Bundle Dispersion

In order to ensure that tendencies of an entire group are collected and not just the quirks of a particular author, a dispersion threshold is required. Dispersion is handled in a number of different ways in existing research. For example, Chen and Baker (2010); Ädel and Erman (2012) require each bundle to appear in at least three separate texts; and Biber et al. (1999, 2004); Cortes (2004); Pan et al. (2016) require each bundle to appear in at least five

---

[4]https://github.com/Hipo/university-domains-list

[5]https://www.familysearch.org/

| Category | Subcategory | Example |
|---|---|---|
| NP-based | NP with of-phrase fragment | *the performance of the* |
| | NP with other post-modifier fragment | *the research on the* |
| | Other NP | *more and more attention* |
| PP-based | PP with embedded of-phrase | *in the form of* |
| | Other PP fragment | *with respect to the* |
| VP-based | copula be + NP/adjective phrase | *is one of the* |
| | VP with active verb | *play an important role* |
| | VP with infinitive verb | *to better understand the* |
| | VP with passive verb | *can be used to* |
| | beginning with past participle | *based on the above* |
| Clause-based | PP + copula be | *of this thesis is* |
| | NP + copula be | *this thesis is to* |
| | Anticipatory it + copula be + adjective phrase | *it is possible to* |
| | Anticipatory it + passive verb + that | *it is found that* |
| | NP/complementizer + passive verb | *little is known about* |
| | NP + active verb | *this thesis focuses on* |
| | NP + active verb + that | *we find that the* |
| Conjunctions | | *As well as the* |

Table 8: Lexical bundle categories with examples, as outlined in Lu and Deng (2019).

separate texts. Due to the imbalance of our five different corpora and the shorter length of each of our texts in comparison to previous works, we require each bundle to appear in at least one percent of abstracts in the corpus of interest.

### B.1.4 Handling Overlapping Bundles

After collecting our list of four-word bundles, there is another processing step that is required and that is to handle overlapping bundles. For instance, in the China corpus there are two overlapping bundles *we propose a novel* and *propose a novel method*. In this case, these two bundles are in fact part of a five-word bundle *we propose a novel method*. In order to avoid inflating the bundle counts, we subsume the later bundle into the former, becoming *we propose a novel (method)*. In deciding which of the overlapping bundles to preserve, we keep the bundle with the highest instance count and discard the bundle with the lower count. In the China corpus, *we propose a novel* occurred more frequently than *propose a novel method* and thus was the bundle that was retained.

## C Appendix

### C.1 Publication Venue Specific Results

### C.1.1 Lexical Complexity

| Publication Venue | Location | Token-Type Ratio | Chain Length |
|---|---|---|---|
| | China | 0.62* | 1.36 |
| | United States | 0.64 | 1.38 |
| ACL | Germany | 0.64 | 1.35 |
| | Japan | 0.62* | 1.32* |
| | India | 0.61* | 1.30* |
| | China | 0.62* | 1.35* |
| | United States | 0.64 | 1.40 |
| EMNLP | Germany | 0.64 | 1.35* |
| | Japan | 0.62* | 1.31* |
| | India | 0.63 | 1.31* |
| | China | 0.60* | 1.36* |
| | United States | 0.63 | 1.39 |
| arXiv | Germany | 0.63 | 1.37 |
| | Japan | 0.60* | 1.34* |
| | India | 0.60* | 1.36* |

Table 9: Token-Type Ratio (calculated as unique tokens divided by total tokens) and Average Lexical Chain Length for each publication venue. Values that are statistically significant when compared with the United States corpus are indicated with * (p-value of 0.05).

## C.1.2 Morphological

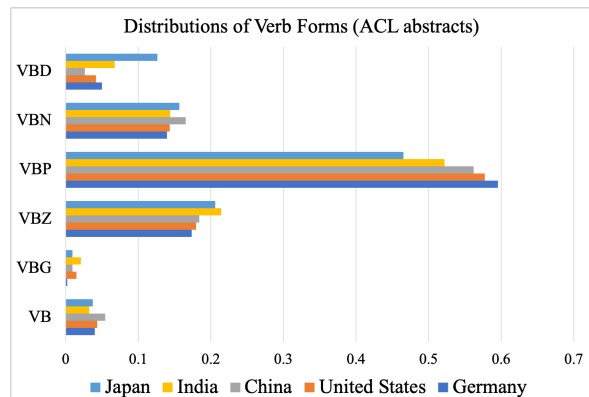

(a) Verb form distributions for abstracts published in ACL

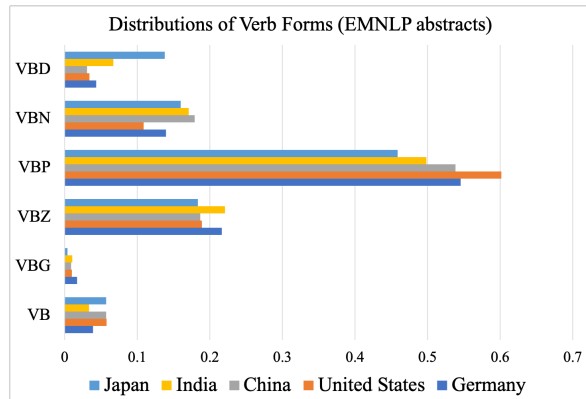

(b) Verb form distributions for abstracts published in EMNLP

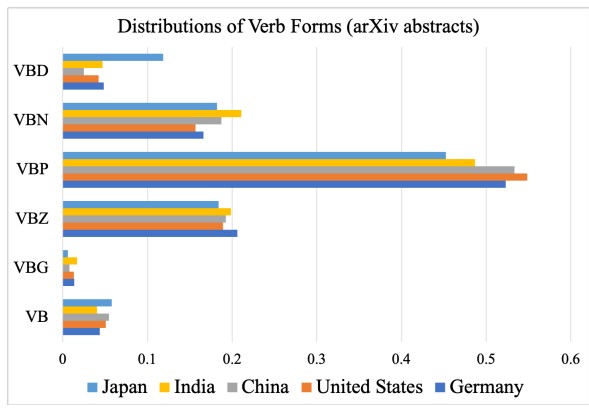

(c) Verb form distributions for abstracts published in arXiv

Figure 2: Distribution of Verb Forms for each corpus in each publication venue: *VBD* denotes past tense form (e.g., *they thought*), *VBN* past participle (e.g., *a sunken ship*), *VB* a verb's base form (e.g., *think*), *VBZ* third-person singular present (e.g., *she thinks*), *VBP* non-third person singular present (e.g., *I think*) and *VBG* gerund (e.g., *thinking is fun*).

## C.1.3 Syntactic Complexity

| Pub Venue | Location | #NP Mod | Cl / Sent | PT Dep | Sent Len |
|---|---|---|---|---|---|
| ACL | China | 1.74* | 1.87* | 6.26* | 24.29* |
| | US | 1.61 | 1.99 | 6.69 | 25.46 |
| | Germany | 1.59 | 2.00 | 6.71 | 25.19 |
| | Japan | 1.65 | 1.91 | 6.59 | 24.22* |
| | India | 1.73* | 1.66* | 6.51* | 23.66* |
| EMNLP | China | 1.74* | 1.89* | 6.25* | 24.18* |
| | US | 1.60 | 2.10 | 6.86 | 26.17 |
| | Germany | 1.64 | 1.88* | 6.71* | 24.33* |
| | Japan | 1.67* | 1.91* | 6.51* | 23.91* |
| | India | 1.76* | 1.62* | 6.61* | 23.32* |
| arXiv | China | 1.75* | 1.87* | 6.24* | 24.71* |
| | US | 1.63 | 2.04 | 7.01 | 26.50 |
| | Germany | 1.64 | 1.86* | 6.62* | 25.17* |
| | Japan | 1.67* | 1.93* | 6.49* | 24.98* |
| | India | 1.68* | 1.70* | 6.67* | 24.19* |

Table 10: Syntactic Complexity Measures for each publication venue in the dataset (*#NP Mod* is the average number of noun-phrase modifiers per noun phrase, *Cl / Sent* is the average number of subordinating clauses per sentence, *PT Dep* is the average parse tree depth for a sentence, and *Sent Len* is the average number of tokens per sentence). Values that are statistically significant when compared with the United States corpus are marked * (p-value of 0.05).

## C.1.4 Cohesion

| Publication Venue | Location | Connectors per Sentence |
|---|---|---|
| ACL | China | 1.88 |
| | United States | 1.95 |
| | Germany | 1.92 |
| | Japan | 1.77* |
| | India | 1.82 |
| EMNLP | China | 1.88* |
| | United States | 2.00 |
| | Germany | 1.99 |
| | Japan | 1.72* |
| | India | 1.75* |
| arXiv | China | 1.90* |
| | United States | 1.98 |
| | Germany | 1.99 |
| | Japan | 1.87* |
| | India | 1.82* |

Table 11: Average number of discourse connectors for each publication venue in our dataset. Statistically significant values are denoted with * (p-value set to 0.05) when compared with the United States corpus.

## D Appendix

In this section, we outline the discourse connector categories and sub-categories that are adopted in our cohesion analyses. There are a total of eight categories: *additive*, *apposition*, *consequential*, *com-*

| Category | Description | Sub-categories | Examples |
|---|---|---|---|
| Additive | to add new information | Equative
Reinforcing | *equally*
*furthermore* |
| Apposition | to give examples or explanation | Exemplification
Restatement | *for example*
*in other words* |
| Consequential | to give cause and effect relationships | Causative
Resultive
Conditional | *due to*
*because*
*whether* |
| Comparison | to point out or imply resemblances between objects | | *like* |
| Contrastive | to introduce surprising or contrasting information | Antithetic
Concessive
Reformulatory
Replacive | *then*
*still*
*or better*
*rather* |
| Clarification | to clarify a point of view, attitude or statement | Equative
Emphasizing
Generalization | *indeed*
*in fact*
*generally* |
| Sequential | to list main points and signal the sequence | Ordering
Timing
Transitional | *first*, *then*
*after*, *immediately*
*by the way* |
| Summative | to conclude or sum up preceding information | | *overall* |

Table 12: Discourse connector categories and sub-categories, as outlined in Kalajahi et al. (2017).

*parison*, *contrastive*, *clarification*, *sequential* and *summative*. These categories are further divided into the following sub-categories: *equative*, *reinforcing*, *exemplification*, *restatement*, *causative*, *resultive*, *conditional*, *antithetic*, *concessive*, *reformulatory*, *replacive*, *equative*, *emphasizing*, *generalization*, *ordering*, *timing* and *transitional*.