# OpenReview forum: "Addressing Linguistic Bias through a Contrastive Analysis of Academic Writing in the NLP Domain"
_EMNLP/2023/Conference — EMNLP 2023 Main_

### Official Review · Reviewer_5giR · 2023-07-30

**Soundness:** 2

**Excitement:**

4: Strong: This paper deepens the understanding of some phenomenon or lowers the barriers to an existing research direction.

**Paper Topic And Main Contributions:**

This paper provides a genre analysis of abstracts of research articles in the NLP field and some pieces of advice to make their usage of English closer to native speakers' one.
Different from many other similar past papers, the paper presents the comprehensive analyses from four different viewpoints: lexical, morphological, syntactic, and cohesive perspectives.
Some of the findings are consistent with some previous EAP work. The findings are as follows:

- Lexical diversity of cn/in/jp is lower than us/de
- Lexical bundles appear more frequently in cn/in/jp/de than us, which is consistent with Hyland (2008a) and others
- Past tense is more used in jp than others
- First-person singular present verbs are less used in jp than others
- Complex phrases are used in in/cn/jp than us, which is partly consistent with Lu and Ai (2015)
- More clauses are used in us than others.
- Discourse connectors are less used in us/de than cn/in/ja, which is partly consistent with Wang (2011)
- "Firstly" appears in cn ten times more than us.
- "Besides" also frequently appears in cn, which is consistent with Yeung (2009)
- "Hence" is overly used in de/in/ja

Methodologies:
- lexis: calculating unique word ratio, using wordnet to identify synonyms/hypernyms
- lexical bundles: 4-grams extracted
- morphology: using spaCy to tag POS
- cohesion: parse-tree

The authors recommend that ACL's guidelines should include some useful expressions.

Also, they claims that many aspects of writing give indication as to whether an author is a native English speaker or not.

**Questions For The Authors:**

1. How did the authors address overlapping bundles? In Sec. B.1.4, "we propose a novel" is selected rather than "propose a novel method", but how about "a novel method for"? What kind of algorithm was applied to this selection?
2. In l.197, it is said that Japanese writers use a lower proportion of first-person singular present verbs. I wonder whether first-person singular present verbs are used in *ACL papers because I have rarely seen them. Could it be possible to raise some examples or number of occurrence?
3. For syntactic analysis what kind of tool did the authors used?

(all answered)

**Reasons To Accept:**

This work presents comprehensive analyses of English usage in *ACL abstracts, in that it includes multiple linguistic perspective and writers' native languages unlike previous work in English for Academic Purposes.

It reveals the differences of English usage among native languages of authors of research articles quantitatively.

**Reasons To Reject:**

- The authors give reasons why the NLP field was targeted: its value in the NLP community, where computational methods are used to study language and linguistic expression is seen as important, which I think does not make sense. As authors mention in the paper, there are many papers working on genre analysis of research articles in different domains. As scientific research, more generalised work is preferred because many discipline-specific (but not comprehensive) studies have been published such as Cunningham (2015) for mathematics and Esfandiari & Barbary (2017) for psychology.
- So, even if focusing only on the NLP domain, comparison with previous findings should be conducted to show the findings are universal or NLP-specific. The paper does not fully compare their findings with past work.

- The work is only focused on abstracts. I do not find any technical/scientific reasons for this. There exists a full-text corpus of ACL Anthology (e.g. ACL Anthology Sentence Corpus), and the methodologies the authors adopted are computer-based ways (counting words, pos taggings, etc.) that do not require much manual labour or many of GPUs. Analyses of abstracts are okay, but in my opinion, it is difficult to recommend something about writing an article based only on the abstract analysis.
- The authors claim that lexical diversity in cn/in/ja is less than that in us/de, but there is only 0.02 point difference, which amounts to about 2-3 words in each abstract. There may be preferences for singular nouns or specific prepositions.

- Although in the past many EAP studies used four-word lexical bundles for analysis, but as Swales (2019) criticises, they contain non-sense bundles like "and of the". Simpson-vlach & Ellis (2010) adopted mutual information, and several methods to extract lexical bundles have been proposed such as Liu et al. (2016) and Brooke et al. (2017). The method should be more carefully considered.

- After reading the review by Nr5L, I find the suggestion in Section 5 goes against recent ACL's efforts. In recent conferences reviewers have been asked not to reject papers for poor English. Academic writing and writing assistance are important but forcing non-native authors to use/not use specific wording unfairly impose a burden.

**Reproducibility:**

4: Could mostly reproduce the results, but there may be some variation because of sample variance or minor variations in their interpretation of the protocol or method.

**Reviewer Confidence:**

4: Quite sure. I tried to check the important points carefully. It's unlikely, though conceivable, that I missed something that should affect my ratings.

---

> ### Author Rebuttal · Authors · 2023-08-28
>
> Thank you for your comments on our work. We address your concerns and questions as follows:
>
> __R1:__ Reasons why the NLP field was targeted.
> __Response:__ In terms of performing our analysis in the NLP domain as opposed to universal domain, we believe that there are certain characteristics of writing that are specific to each domain. For example, in the NLP domain, writers are often required to write with a high degree of technical detail, often outlining algorithm details; this may impact the syntactic complexity and the sentence structures that are used for instance. We felt that performing our analysis on universal domain data might reduce the impact of some of the phenomena observed in our study. Additionally, regarding why the NLP domain was selected specifically, it is the field we work in and are engaged in, and we are committed to contributing to help the NLP community improve and thrive.
>
> __R2:__ It is difficult to recommend something about writing an article based only on the abstract analysis.
> __Response:__ In our study, we chose to analyze paper abstracts due to the fact that abstracts are clean and often carefully written; additionally, they are all written with the same function – that is to summarise the contributions of the paper. These characteristics make abstracts easily comparable and thus well-suited for a comparative analysis. Performing an analysis of whole papers is also of value, however, with many potential new insights to be gained, and thus we reserve this for future work.
>
> __R3:__ The authors claim that lexical diversity in cn/in/ja is less than that in us/de, but there is only 0.02 point difference, which amounts to about 2-3 words in each abstract.
> __Response:__ While the lexical diversity scores do not appear to be hugely different between the cn/in/ja corpora and the us/de corpora, we support our results with a pairwise t-test, comparing each corpus with the US corpus. In this case, each sample is the score from one abstract. The p-value is set to 0.05. The results are as follows, with statistically significant results denoted by *.
>
> __Table__ __2:__
> |			| Token-Type Ratio	| Chain Length |
> | --------------- | :---------------------: | :---------------: |
> | China |	*	| * |
> | Germany |		| * |
> | Japan |	*	| * |
> | India |	*	| * |
>
> While lexical diversity could be down to preferences for singular nouns or certain prepositions, we believe that our lexical-chain length analysis supports the hypothesis that the higher token-type ratio score in the US corpus is as a result of more synonym, hypernym and hyponym usage.
>
> __R4:__ In the past many EAP studies used four-word lexical bundles for analysis, but as Swales (2019) criticises, they contain non-sense bundles like "and of the".
> __Response:__ With regards to using four-word bundles, we found that this was the most beneficial bundle size to obtain useful insights, as three-word bundles extract many common non-sense bundles, as highlighted in the review; and five-word bundles are too infrequent. There are some extra post-processing steps we took to ensure that our bundles were informative. For instance, any non-sense bundles were removed by hand (these were very infrequent), as were any bundles that contained technical language or jargon that was relevant only for the domain in which it appears (e.g., ‘aspect based sentiment analysis’). Other methods for extracting lexical bundles such as mutual information are definitely worth exploring, and we reserve this for future research.
>
> __QA__: How are overlapping bundles handled?
> __Response__: In the case of overlapping bundles, when choosing which of ‘we propose a novel’ and ‘propose a novel method’ to record, the bundle with the highest instance count is chosen. For ‘a novel method for’, it will also be recorded as a separate bundle if it is not part of another overlap with a higher instance count (e.g., if ‘propose a novel method’ has a higher instance count, then ‘a novel method for’ will not be recorded). We will add the algorithm details in our revised version.
>
> __QB__: Any examples or instance counts for first-person singular?
> __Response__: Below we have included a few examples of first-person singular instances extracted from our dataset:
>
> “I present an end-to-end probabilistic model …”
> “I collect a novel dataset …”
> “I compare the model-based …”
>
> In addition, we have included the most common first-person singular verbs for each corpus and the proportions they occupy within each corpus.
>
> |			| 1	|  2 |  3 | 4 |
> | --------------- | :---: | :---: | :---: | :---: |
> | China |	propose (0.21)	| show (0.13) | demonstrate (0.05) | introduce (0.03) |
> | United States |	show (0.10)	| propose (0.08) | present (0.06) | demonstrate (0.05) |
> | Germany |	show (0.11)	| propose (0.08) | present (0.07) | demonstrate (0.04) |
> | Japan |	propose (0.19)	| show (0.14) | demonstrate (0.06) | introduce (0.04) |
> | India |	propose (0.13)	| show (0.08) | present (0.08) | demonstrate (0.04) |
>
>
> __QC__: Which tool was used for syntactic analysis?
> __Response__: For our syntactic analyses, a range of tools were used. For word and sentence tokenization, we used nltk, for tree parsing, we used the CoreNLP package from Stanford, and for clause extraction, we used clausie (https://spacy.io/universe/project/spacy-clausie).

---

### Official Review · Reviewer_QejG · 2023-08-05

**Typos Grammar Style And Presentation Improvements:** p.8, bus -> but
**Soundness:** 5

**Excitement:**

4: Strong: This paper deepens the understanding of some phenomenon or lowers the barriers to an existing research direction.

**Missing References:**

N/A

But (First et al.2019) should be (Education First, 2019)

**Paper Topic And Main Contributions:**

This is an excellent paper, which addresses a very important problem, that of linguistic bias in academic reviewing. The issue of how reviewers can be influenced by the perceived native language of authors and by prejudices against non-native English speakers/writers in an environment dominated by native English speakers must be tackled and this is a very good start. The authors not only provide examples of divergences between the lexical choices and grammatical constructions of authors of different nationalities, but also make useful suggestions for publishers and journals and conference editors. The linguistic analyses are detailed and well-documented, with good examples.
The data should be replicable when the authors make their data available.



**Questions For The Authors:**

Please make it clear whether you are making general statements about L1/L2 or about native English vs. non-native English. The fact that this not always done by the authors you cite doesn’t make it more acceptable,
It would be important to discuss the linguistic supremacy of US-based academic writing and point out that English is not monolithic. For instance, why the “native” corpus is labelled “US”.  We might also expect differences between US/UK/Australian usage. If not, why not?

Why not suggest that US-based writers adopt international usage? For instance, you note that the Japanese corpus shows “under-use of first-person present tense (only 45.5% of main verbs are first-person present, in comparison to 56% in the United States corpus).” This could also be considered as over-use of first person present by US writers.  Who decides the US usage has to be the norm?

The recommendations in section 5 will be very useful for editors, reviewers, journals and conferences. But why insist that authors should display a range of variations? Formulaic language (lexical bundles) can actually be useful for both writers and readers.

**Reasons To Accept:**

See above

**Reasons To Reject:**

None regarding the content of the paper. But the text must be made clearer regarding the terms “native speakers” and non-native speakers” and whether specific remarks apply only to writers who do not have English as a first language, or to any writing in a non-native language.
The authors must ensure that everywhere the terms "native" and "non-native" are used it is clear what language they are referring to. Almost everyone is a native speaker of at least one language and the generic term "non-native speaker", as often used to refer to non-native English speakers, is insulting and demeaning.
This is especially necessary in the abstract ("non-native authors") and the conclusion ("native or non-native"), but also throughout the text. There are many places discussing research in second language writing where it is not clear whether it is actually EFL (English as a foreign language) or L2 (any second language) that is being discussed.


**Reproducibility:**

5: Could easily reproduce the results.

**Reviewer Confidence:**

4: Quite sure. I tried to check the important points carefully. It's unlikely, though conceivable, that I missed something that should affect my ratings.

---

> ### Author Rebuttal · Authors · 2023-08-28
>
> Thank you for your comments on our paper regarding its importance to fairness in academic reviewing. Regarding your concerns and questions, we address them as follows:
>
> __QA__: Distinction between L1/L2 and native English / Non-native English
> __Response__: We agree that where ‘native’ and ‘non-native’ are used it should be clearly defined as to which language is being referred to. We will clarify each usage of these terms in the revised version of this paper. Regarding the US corpus being adopted as the native corpus, we will also add clarification that while US English is dominant in academic publishing, it is not the only variation of English and thus our analyses do not necessarily extend to other locations where English is also a first language. It is possible that other varieties of English (UK, AUS, etc.) could exhibit differences in comparison to the US, particularly with word choices. For future research, it would certainly be of value to perform analyses across more locations where English is used as a first language.
>
> __QB__: Why not suggest US adopt international standards for academic writing?
> __Response__: The suggestion for US writers to adopt international standards is something that could be considered as an ideal scenario, and is something that could potentially be explored in the long term. One of the barriers that our work attempts to make progress on is to aid writers according to existing publishing conventions, whereby a culture has developed of nativelike English being considered prestige and many publishing venues often adopting US-based style guides.
>
> __QC__: Range of variations suggestion in Section 5
> __Response__: Lexical bundles can certainly be of benefit to writers. However, through our analysis, we have identified that there are certain bundles that are used more frequently by writers from certain geographic locations. High frequency use of lexical bundles is likely to impact how the reader evaluates the quality of the writing, as text may feel repetitive and perhaps even tedious. It is likely the case that the degree of diversity that is observed in the US corpus is a factor in how the quality of a text is perceived. Therefore, bundle diversity is likely to be useful for non-native English writers, too.

---

### Official Review · Reviewer_Nr5L · 2023-08-06

**Soundness:** 3

**Ethical Concerns:**

Yes

**Excitement:**

4: Strong: This paper deepens the understanding of some phenomenon or lowers the barriers to an existing research direction.

**Justification For Ethical Concerns:**

Section 5 needs to pass the ethical test.

**Paper Topic And Main Contributions:**

This paper presents an exploratory study of linguistic variation in academic writing across geographical locations. Authors have explored linguistic variation across four dimensions, (i) Lexical, (ii) Morphological, (iii) Syntax, and (iv) Cohesion. The geographical locations they focused on are (i) China, (ii) US, (iii) Germany, (iv) Japan, and (v) India. For their study, they considered abstracts from two prominent NLP research venues, ACL and EMNLP, for papers published between 2018-2021 and 2017-2021, respectively. In addition, they considered abstracts from Arxiv papers tagged with ‘Computation and Language’ published between 1st Jan 2015 to 24th April 2022. They followed various heuristics (Appendix A) to identify the origin country of the considered papers (First author, to be precise), which sounds robust.

Various well-known approaches, such as token-type ratio, lexical chan length, lexical bundles per million words etc., were calculated to showcase the linguistic difference across the papers from the considered geographical locations. Finally, they have proposed some guidelines for the journal and conference publishers to prepare instructions for non-Native writers to reduce the bias in their writing.

**Reasons To Accept:**

1. The level of detail the authors considered for their study is very impressive. Lots of insights, some of which reiterate or contradict the findings in the literature, were mentioned. In addition to that, numerous new linguistic insights were presented, which will be a delight to read by a linguist/computational linguist.

2. The heuristics considered in this study for paper selection are pretty robust. This ensures one of the founding pillars for quality dataset creation for this study.

3. This study will interest to a broader NLP community especially those who like the linguistic aspect in the NLP.

**Reasons To Reject:**

There are several fundamental weaknesses I find in this study.

1. Authors have showcased many results showcasing the linguistic difference between native and non-native writers. Most of the time, the numerical deviation in the values is not very high. In this scenario, examining statistical significance is a primary requirement, and unfortunately, the authors have not done that for a single outcome. WIth out this, I am taking the results with a pinch of salt.

2. Authors have not given any statistical distribution on paper venues. How many papers are from ACL and EMNLP vs Arxiv? Everyone knows Arxiv papers are not peer-reviewed and are hardly considered formal publications. My intuition is that it's the arxiv papers that showcase this bias. If this is the case, I would not like to consider the results presented in this paper as significant. Also, I have severe reservations about including of Arxiv papers in this study. Most of Arxiv's papers are junk and are not considered as severe writing at all.

3. Most of the recommendations mentioned in section 5 are guidelines for publishers to encourage non-native users to use different writing patterns by showcasing examples. This advisory is very racist and discriminatory towards non-native writers. Instead, authors should propose guidelines for reviewers on ‘Look into the contributions, not writing patterns’. Research papers stand for their technical and empirical contributions. They are not literary contributions for which they should be advised on ‘how to write’.  Also, in the era of AI tools like Grammarly, why do the authors think that there is a need for such recommendations?

**Reproducibility:**

3: Could reproduce the results with some difficulty. The settings of parameters are underspecified or subjectively determined; the training/evaluation data are not widely available.

**Reviewer Confidence:**

4: Quite sure. I tried to check the important points carefully. It's unlikely, though conceivable, that I missed something that should affect my ratings.

---

> ### Author Rebuttal · Authors · 2023-08-28
>
> Thank you for your appreciation regarding the level of detail, comments regarding the robustness of our dataset creation procedure and the interest of our paper to the NLP community. We value your concerns regarding (1) the statistical significance of the results, (2) distribution of paper venues, and (3) guidelines outlined in Section 5, and address them as follows:
>
> __1__: For instances where numerical deviation between different corpora appears to be small, we include statistical significance results from a pairwise t-test for each corpus compared with the United States corpus, with the p-value set at 0.05 (statistically significant results denoted with *) and individual samples being either for each abstract (e.g., token-type ratio) or for each sentence (e.g., clauses per sentence):
>
> &nbsp;
>
> __Table__ __2:__
> |			| Token-Type Ratio	| Chain Length |
> | --------------- | :---------------------: | :---------------: |
> | China |	*	| * |
> | Germany |	| * |
> | Japan |	*	| * |
> | India |	*	| * |
>
> &nbsp;
>
> __Table__ __5:__
> |			| #NP Mods	| #Clause per sent | #Parse Tree Depth | #Sent Length |
> | --------------- | :--------------: | :------------------: | :-------------------: | :---------------: |
> | China |	*	| * | * | * |
> | Germany |	| * | * | * |
> | Japan |	*	| * | * | * |
> | India |	*	| * | * | * |
>
> &nbsp;
>
> __Table__ __6:__
> |			| #Logical Connectors per Sentence |
> | --------------- | :---------------------------------------: |
> | China |	*	|
> | Germany |	|
> | Japan |	*	|
> | India |	*	|
>
> &nbsp;
>
> In the revised version of this paper, we will include statistical significance results where relevant.
>
> &nbsp;
>
> __2__: Regarding venue distributions, we list the paper counts in our dataset for each venue:
>
> | Venues | #Abstracts |
> | ----- | :-----: |
> | ACL | 999 |
> | EMNLP | 1289 |
> | ARXIV | 5180 |
>
> While we appreciate the concern that arxiv papers do not exhibit the same writing quality of those from ACL and EMNLP, we add that the distributions of papers from the different geographical locations are almost identical across the venues; therefore, we believe it to be unlikely that differences between the different locations can be attributed to the different publication venues.
>
> Additionally, we have also carried out a number of experiments on papers from each individual publication venue, and have found the phenomena observed to be consistent with what is published in our paper for all three venues combined. For instance, the usage of logical connectors from each venue and for each country are displayed in the following table:
>
> &nbsp;
>
> __#Logical Connectors per Sentence__
> |			| ACL | EMNLP | ARXIV |
> | --------------- | :----: | :-------: | :------: |
> | China |	1.88	| 1.88 | 1.90 |
> | United States | 1.95 | 2.00 | 1.98 |
> | Germany | 1.92 | 1.99 | 2.00 |
> | Japan | 1.77 | 1.72 | 1.86 |
> | India |	1.82	| 1.75 | 1.81 |
>
>
> __3__: Regarding the concern that the recommendations outlined in Section 5 may be interpreted as discriminatory, our intention is to provide assistance to non-native speakers of English based on the findings of our analysis. We will make some wording changes to Section 5 in our revised version in order to address this issue. The suggestion for reviewers to focus on the content rather than the writing patterns is good and is the ideal scenario for academic publishing; this should certainly also be recommended by publication venues. As an additional note, however, research has shown that reviewers often judge work more harshly when it is less nativelike, even when the communicability is unaffected by non-native writing patterns (examined by work cited in the second paragraph of the introduction). Thus, we believe that providing writers with ways to assist them in being evaluated on the quality of their research is highly valuable to the research community.

---

### Meta-Review · Area_Chair_STxc · 2023-09-18

**Recommendation:** 5

**Metareview:**

This paper presents a linguistical analysis of how NLP paper abstracts vary by nationality of the first author. The topic addressed, linguistic bias, is an important one, and overall, the reviewers are positive. However, the reviewers also raise a number of concerns related to presentation that undermine the impact of the authors' work, such as referring to "native speakers" rather than "native English speakers". Some of the assumptions made in the text (i.e. that Indian authors are not native English speakers) seem easily avoidable by discussing between-country differences rather than categorizing authors from specific countries as native or non-native English speakers. Bias towards differences between, for example, Indian English and American English are just as pertinent to the broader discussion as bias against non-native English speakers.

In summary, the reviewers seem to agree that the topic is important and the work is technically sound; one reviewer even recommends the paper for an award. Yet all reviewers also agree that some of the paper's language needs revision.

---

### Decision · Program_Chairs · 2023-10-07

**Decision:**

Accept-Main

**Comment:**

This paper presents a linguistical analysis of how NLP paper abstracts vary by nationality of the first author. The topic addressed, linguistic bias, is an important one, and overall, the reviewers are positive. However, the reviewers also raise a number of concerns related to presentation that undermine the impact of the authors' work, such as referring to "native speakers" rather than "native English speakers". Some of the assumptions made in the text (i.e. that Indian authors are not native English speakers) seem easily avoidable by discussing between-country differences rather than categorizing authors from specific countries as native or non-native English speakers. Bias towards differences between, for example, Indian English and American English are just as pertinent to the broader discussion as bias against non-native English speakers.

In summary, the reviewers seem to agree that the topic is important and the work is technically sound; one reviewer even recommends the paper for an award. Yet all reviewers also agree that some of the paper's language needs revision.